# GLP1 Exerts Paracrine Activity in the Intestinal Lumen of Human Colon

**DOI:** 10.3390/ijms23073523

**Published:** 2022-03-24

**Authors:** Carme Grau-Bové, Carlos González-Quilen, Giulia Cantini, Patrizia Nardini, Beatriz Espina, Daniele Bani, Ximena Terra, MTeresa Blay, Esther Rodríguez-Gallego, Michaela Luconi, Anna Ardévol, Montserrat Pinent

**Affiliations:** 1MoBioFood Research Group, Department of Biochemistry and Biotechnology, Universitat Rovira i Virgili, 43007 Tarragona, Spain; carme.grau@urv.cat (C.G.-B.); carlosalberto.gonzalez@alumni.urv.cat (C.G.-Q.); ximena.terra@urv.cat (X.T.); mteresa.blay@urv.cat (M.B.); esther.rodriguez@urv.cat (E.R.-G.); montserrat.pinent@urv.cat (M.P.); 2Endocrinology Unit, Department of Experimental and Clinical Biomedical Sciences, University of Florence, 50139 Florence, Italy; giulia.cantini@unifi.it; 3Research Unit of Histology & Embryology, Department of Experimental and Clinical Biomedical Sciences, University of Florence, 50139 Florence, Italy; patrizia.nardini@unifi.it (P.N.); daniele.bani@unifi.it (D.B.); 4Servei de Cirurgia General i de l’Aparell Digestiu, Hospital Universitari Joan XXIII, 43005 Tarragona, Spain; bespina84@gmail.com; 5Institut d’Investigació Sanitària Pere Virgili (IISPV), 43005 Tarragona, Spain

**Keywords:** GLP1, PYY, apical secretion

## Abstract

GLP1 produced in the upper part of the gut is released after food intake and acts by activating insulin secretion, but the role of GLP1 in the colon, where it is predominantly produced, remains unknown. Here we characterized the apical versus basolateral secretion of GLP1 and PYY and the paracrine mechanisms of action of these enterohormones in the human colon. We stimulated human colon tissue in different ex vivo models with meat peptone and we used immunofluorescence to study the presence of canonical and non-canonical receptors of GLP1. We found that PYY and GLP1 are secreted mainly at the gut lumen in unstimulated and stimulated conditions. We detected DPP4 activity and found that GLP1R and GCGR are widely expressed in the human colon epithelium. Unlike GLP1R, GCGR is not expressed in the lamina propria, but it is located in the crypts of Lieberkühn. We detected GLP1R expression in human colon cell culture models. We show that the apical secretion of PYY and GLP1 occurs in humans, and we provide evidence that GLP1 has a potential direct paracrine function through the expression of its receptors in the colon epithelium, opening new therapeutic perspectives in the use of enterohormones analogues in metabolic pathologies.

## 1. Introduction

Gut hormones are responsible for regulating important metabolic pathways in the body such as nutrient uptake, energy homeostasis, gut motility, food intake regulation, and cell differentiation [1]. Among other signals, the glucagon like peptide 1 (GLP1) and peptide YY (PYY) are released when nutrients reach the intestinal lumen and act as satiety hormones through vagal afferent nerves activated in the intestinal mucosa. Moreover, both GLP1 and PYY slow gut motility and delay gastric emptying [2]. Unlike PYY, GLP1 secretion in the proximal intestine controls glycaemia because of its incretin effect. L-cells, i.e., GLP1 and PYY producing cells, are increasingly abundant towards the distal intestine, and concentrations are highest in the colon [3]. It has been hypothesized that the function of L-cells in the colon is different from the function of those in the proximal intestine [4]. It is believed that the main function of GLP1 secretion by colonic L-cells is to slow gut motility [5,6]. While PYY is known to slow colon motility through the Y2 receptors, it is not known whether the GLP1 mechanism of action in the colon is different from the mechanism in the proximal intestine.

In recent years, our understanding of GLP1′s action and functions has changed considerably. Many non-insulinotropic functions of GLP1 have been found, most of which have been attributed to its cleaved form, GLP1 (7–36) [7], initially believed to be inactive. It seems to function independently of GLP1R, the canonical receptor of the full form GLP1 (7–36), as its binding affinity is extremely low [8]. It has been postulated that several receptors recognize GLP1 (9–36), including CD36/fatty acid transporter [9] and the receptors for the members of the proglucagon gene family to which GLP1R belongs, such as the glucagon receptor (GCGR) [10]. In a recent study by Guida et al. [11], GLP1 (9–36) was shown for the first time to act through GCGR to inhibit glucagon secretion from pancreatic α-cells, while GLP1 (7–36) had the same effect through GLP1R.

Stevens et al. [12] recently identified luminal GLP1 and PYY secretion. This finding contrasts sharply with the widely accepted idea that enteroendocrine cells are activated apically and react by secreting enterohormones from the basolateral membrane and making them available for the vagal and afferent nerves in the lamina propria [2,13]. Along the same lines, exogenous GLP1 has also been identified in the lumen of rat intestines. It was very recently shown to be released by *Lactobacillus paracasei*, present in the microbiota, and postulated to act via GLP1R expressed on some L-cells with a paracrine role [14].

In this paper, we aimed to provide further insight into apical versus basolateral secretion of GLP1 and PYY in the human colon and to explore the presence of apical canonical and non-canonical GLP1 receptors in the colon epithelium to understand the importance of apical enterohormone secretion in humans. We stimulated various ex vivo experimental models to test human colon samples with a meat peptone treatment, a common stimulus for enteroendocrine secretions such as GLP1 [15,16] and PYY [17].

## 2. Results

### 2.1. Meat Peptone Stimulates PYY and GLP1 Secretion in Human Colon

The first approach to assess if an enterohormone secretory stimulus such as meat peptone can stimulate the secretome in ex vivo experimental models was to analyze the conditioned media after treatment. Figure 1a,b shows that meat peptone treatment significantly stimulated the secretion of peptide YY (PYY) and glucagon-like peptide 1 (GLP1) from the human colon.

In order to dissect the compartment in which the secretion occurred, we used Ussing chambers, which compartmentalize the apical and basolateral sides of the intestinal samples treated ex vivo. Apical treatment of samples with 50 mg/mL meat peptone resulted in GLP1- but not PYY-stimulated secretion in the basolateral compartment of human colon samples (Figure 1c).

To discard the hypothesis of tissue damage induced by meat peptone treatment, we monitored the effect of different concentrations of meat peptone on the transepithelial electric resistance (TEER) of human colon, which is an index of tissue integrity. We observed that 50 but not 15 mg/mL meat peptone decreased human colon TEER by 30% after a 30-min treatment in the ex vivo model (Appendix A). Therefore, we used the low concentration to subsequently evaluate the effects of meat peptone on apical secretion of enterohormones in Ussing chambers model with human samples.

### 2.2. GLP1 and PYY Are Secreted into the Intestinal Lumen of Human Colon

We then used Ussing chambers to dissect the compartment of secretion and study how apical + basolateral treatment affects GLP1 and PYY secretion from the human colon mucosa.

Meat peptone has no effect on PYY secretion in explants on the basolateral side, but meat peptone significantly increased PYY secretion into the apical side (independently of the side of stimulation) (Figure 2a).

Similarly, 15 mg/mL meat peptone is able to rise GLP1 secretion on the apical side of human colon mucosa (Figure 2b). Interestingly, the basal secretion on the apical side was higher than that on the basolateral side for both GLP1 and PPY (Figure 2a,b).

### 2.3. GLP1 and PYY Are Secreted into the Intestinal Lumen of Human Colon

The first step in determining whether enterohormones exert a function in the apical lumen is to investigate whether their receptors are present on the apical membrane. Focusing on GLP1, we evaluated whether we could detect GLP1R in the apical side of mucosa.

Initially, Western Blot analysis of protein extracts from ascending and descending human colon mucosa showed that they both expressed similar amounts of GLP1R (Appendix A). Immunofluorescence analyses showed an intense GLP1R signal throughout the epithelial layer of the mucosa (Figure 3a). This layer is composed predominantly of colonocytes, but also of goblet cells, which are the mucin-producing cells, and enteroendocrine cells, which are mainly L-cells in the colon. Immunofluorescence microscopy suggested that GLP1R positivity is restricted to colonocytes in the colon epithelium, and is not expressed in L-cells, stem cells or goblet cells (marked with an arrow, Figure 3a). We also detected intense positivity for GLP1R in some cells of the lamina propria (marked with a circle, Figure 3a. Found beneath the lamina propria are the crypts of Lieberkühn or intestinal glands. The negativity of the GLP1R signal in the intestinal glands indicates that the cell types that are also in the epithelium do not express GLP1R. A similar result was obtained working with in vitro models of intestine: (1) a monoculture of CaCO_2_, (2) a co-culture of CaCO_2_ cells with THP1 macrophages, and, finally, (3) a tri-culture of CaCO_2_ with THP1 macrophages and Raji B lymphocytes. All these different in vitro models were grown in cell inserts with membranes cut transversally to resemble mucosal sections with polarized CaCO_2_ cells. We found that GLP1R was abundantly expressed in CaCO_2_ cells in all culturing conditions, but not in the infiltrating cells (Figure 3c–e).

Detection of GLP1R positivity in the colonic epithelial cells raised the question of whether GLP1R was expressed in their apical membrane where lumen stimuli (nutrients, hormones) act. To investigate this, we performed immunofluorescence of GLP1R together with specific staining of compartment markers, using actin staining as a marker of the apical side of epithelial cells [18]. Immunofluorescence staining of human colon with actin revealed that actin clearly marks the apical membrane of the colon epithelium and that it is also widely expressed in all cell types in this layer, the lamina propria, and the crypts of Lieberkühn (Figure 4a). Moreover, co-staining of human colon mucosa with GLP1R and actin showed that the GLP1R signal is located beneath the apical membrane in the epithelial cells, with no colocalization with actin. However, GLP1R co-localized with actin in some cells of the lamina propria (Figure 4b). In addition, PAS staining of mucins revealed that our samples were not covered by mucus, which was present only inside goblet cells (Appendix A).

Actin co-staining was also used as a marker of the apical surface in the in vitro intestinal models. In fact, although THP1 and Raji B express actin diffusely in the cytosol [19,20], only polarized CaCO_2_ cells are expected to clearly show organized actin at the apical edge. Actin staining at the apical surface of CaCO_2_ cells confirmed their polarization in mono- and co-culture conditions. Moreover, GLP1R co-staining with actin filament in CaCO_2_ cells suggests that the GLP1R signal is below the apical surface, in the inside of the cell (Figure 4d,e), as observed in epithelial cells (Figure 4b).

### 2.4. Alternative GLP1 Receptors May Be Responsible for GLP1 (9–36) Sensing in Human Colon Lumen

GLP1 cleavage by dipeptidyl peptidase IV (DPP4) is relevant to understand the role of GLP1 secretion into the intestinal lumen. This protease that cleaves enterohormones and, among others [21], can be located at the apical surface of colonocytes and the vascular endothelium of the capillaries of the lamina propria [22,23]. Since we detected DPP4 activity in the human mucosa of the ascending and descending colon (Figure 5a), we can assume that the GLP1 secreted into the lumen can be cleaved by DPP4 to GLP1 (9–36).

It has recently been suggested that the GLP1 (9–36) cleaved form is active through binding to an alternative receptor [24]. Glucagon receptor (GCGR) has been suggested to mediate the activity of the DPP4-cleaved GLP1 form, at least in preadipocytes [24]. Immunofluorescence staining revealed that GCGR was widely expressed throughout the human colon mucosa. The GCGR signal was localized in the epithelium, like GLP1R. In contrast with GLP1R, however, GCGR was expressed in the crypts of Lieberkühn, but not in the lamina propria (Figure 5b). GCGR and GLP1R immunofluorescence co-staining showed that although the signals did not co-localize, both receptors were expressed together in the same epithelial cells. While the GLP1R signal was localized at the apical edge of colonic epithelial cells, the GCGR signal was stronger at the basal pole of the cells.

Moreover, while only GLP1R was found in the lamina propria, GCGR can be expressed in other cell types in the crypts of Lieberkühn (Figure 5c). Finally, we assessed GCGR expression in the in vitro intestinal co-culture models of CaCO_2_ cells, where we observed widespread expression of GCGR in CaCO_2_ but unclear expression in THP1 cells. When we performed double fluorescence immunostaining of GLP1R and GCGR in the co-culture model, we did not observe a clear localization of these proteins in the same cells (Appendix A).

## 3. Discussion

In the present paper, we have described for the first time how treating colon mucosa with meat peptone affects the luminal secretion of PYY and GLP1, and whether the stimulation was more effective on the apical or the basolateral side of the intestine. In human colon, we observed that in basal conditions and after meat peptone treatment both PYY and GLP1 secretion into the luminal (apical) compartment was higher than into the basolateral compartment. We also found that, in comparison with control conditions, meat peptone only significantly increases PYY and GLP1 in the luminal compartment and that treatment other than just apical stimulation (apical + basolateral stimulation) does not have a different outcome. Thus, meat peptone apically stimulates PYY and GLP1 secretion, which also occurs apically.

Luminal secretion of PYY and GLP1 from human colon was first reported by Stevens et al. [12], who used another ex vivo vector model known as InTESTine. They evaluated the response of human colon to apical treatment with 12.5 mm rebaudioside A or 2.5% casein. In response to the casein treatment, GLP1 was only stimulated into the lumen while PYY secretion was higher into the basolateral compartment than into the lumen. These results are similar to ours only in the case of GLP1, since in response to meat peptone GLP1 and PYY secretion occurred apically. Stevens et al. showed similar results with the rebaudioside treatment. Unlike meat peptone, rebaudioside did stimulate GLP1 secretion in the basolateral side, although apical secretion was still higher in both control and treated conditions.

The activity of luminal secreted enterohormones may be influenced by the DPP4 present in the intestinal lumen. DPP4 activity in the intestinal lumen is not only due to intestinal epithelium transmembrane expression of the enzyme [21], but also to microbiota DPP4-like activity, as explained by Olivares et al. [25]. This is confirmed by the fact that DPP4 activity is greater in fecal material in microbiota-colonized mice than in germ-free mice (GFM). Olivares et al. hypothesized that through DPP4 activity, microbiota could affect the host metabolism by protein digestion in the lumen and by regulation of enterohormone secretion in response to microbial DPP4 that has crossed the epithelial intestinal barrier. One finding that supports this hypothesis is that DPP4 KO mice showed blood DPP4 activity of unknown origin, which could be explained by microbial DPP4 that has crossed the epithelial intestinal barrier. Our findings provide novel biological relevance to this microbiota–host interaction through direct microbiota DPP4 activity on the luminal PYY and GLP1.

Cleaved GLP1 (9–36) has generally been thought to be inactive since it has no action on the glucose metabolism and a very low binding activity on GLP1R [26]. However, Cantini et al. [10] showed that GLP1 (9–36) has a role in non-pancreatic functions, particularly human preadipocytes, in which in vitro treatments induced inhibition of cell proliferation and differentiation [24]. Since GLP1 (9–36) effects seem to be independent of GLP1R activation, the GLP1 heterodimeric receptor consisting of GLP1R and other receptors such as GCGR has been suggested as alternative to the classic homodimer GLP1R. Accordingly, GCGR has been confirmed to be the GLP1 (9–39) receptor in the pancreas [11]. Here, we have detected for the first time the presence of GLP1R and GCGR on human colon mucosa: GLP1R is expressed in the epithelium and the lamina propria, while GCGR is expressed in the epithelium and the crypts of Lieberkühn. We have also detected GLP1R and GCGR expression in CaCO_2_ colonocytes cocultured with THP1 macrophages and Raji B lymphocytes. The GLP1R and GCGR positivity in the human colon epithelium may suggest that can colonocytes express both receptors. Although we did not observe a clear co-localization of GLP1R and GCGR, they are undoubtedly co-expressed in the same cells, and they might co-localize in stimulated conditions. Our intestinal preparations were not stimulated since they were prepared after being washed in KRB-D-Mannitol. Several studies have shown that GLP1R is located in vesicles in pancreatic-beta cells and translocates to the plasma membrane when stimulated [27,28]. Thus, further studies need to be made with human colon tissue in stimulated conditions to determine whether GLP1R and GCGR form a heterodimer that interacts with GLP1 (9–39). Notably, both GLP1R and GCGR antagonists in adipose precursors have been shown to reverse glucagon inhibitory effects [29].

In addition to colonocytes, the colon epithelium hosts other cell types: endocrine cells and goblet cells. Goblet cells do not express GLP1R because they are found in the crypts of Lieberkühn, which are clearly negative for GLP1R. However, GCGR is more probably expressed in goblet cells, as well as in other cell types in the crypts. The lamina propria is composed of a variety of structural cells and immune cells, nerve endings, and vascular vessels. We observed that several cells of the lamina propria expressed GLP1R: neurons [30], blood vessels [31], intraepithelial lymphocytes [32], and smooth muscular cells [33]. However, in the present paper we did not further characterize their presence in the samples we analyzed. It is still to be clarified whether L-cells, the GLP1- and PYY-producing endocrine cells in the colon epithelium, express GLP1R and GCGR, thus indicating that luminal GLP1 has an autocrine role. However, the expression of GLP1R in the colon epithelium suggests that luminal GLP1 has a regulatory function in this tissue. Several paracrine loops, including GLP1, have been identified. In the small intestine of mice, enterochromaffin cells, endocrine cells which produce 5-HT, are not activated by nutrients in the lumen but by GLP1 secreted by the neighboring cells [34]. In addition, GLP1 indirectly stimulates L-cell proliferation because the release of GLP1 triggered by bile acids leads to serotonin secretion which, in turn, increases the number of L-cells [35]. This also suggests that the GLP1 receptor is present in intestine endocrine cells. Another recent study showed that D-cells, neighboring cells to L-cells in the mouse proximal small intestine, produce somatostatin which inhibits GLP1 secretion [36]. All these findings highlight the paracrine loops involving GLP1 in the proximal part of the intestine. However, in our study, we did not observe GLP1R signal in the crypts of Lieberkühn suggesting that the paracrine role of GLP1 is not mediated by classical GLP1R in the colon. The expression of GCGR but not of GLP1R in some colon cell types reveals a possible novel role for this receptor in mediating cleaved GLP1 activity and for glucagon in human colon mucosa. In fact, it has just been reported that glucagon is released from ileal and colonic human explants [37]. According to this report, glucagon secretion in the intestine is part of normal physiology, and not a compensatory mechanism for disrupted pancreatic secretion, since all donors in the study had a healthy pancreas. Another recent study revealed a paracrine modulation of GCGR in L-cells, since GCGR antagonism increases the L-cell population [38].

Finally, PYY is also a substrate of DPP4. Unlike other DPP4 substrates, its cleaved, PYY (3–36) form has higher affinity than its intact form, PYY (1–36), for one of its receptors, Y2. Conversely, PYY (1–36) binds with similar affinity to its four different receptors, Y1-Y4. The activation of Y2 is associated with the anorectic effects of PYY and delayed gastric emptying, while the activation of Y1 receptor increases intestinal motility and orexigenic effects (see Deacon et al. [23]). The fact that this dichotomic function of PYY depends on DPP4 activity is further confirmed by the effect of DPP4 inhibitors on body weight. While the inhibition of DPP4 increases the levels of GLP1 (7–36), leading to an increase in its anorectic effect, the increased levels of PYY (3–36) counteract weight loss through its orexigenic effect [39]. While both Y1 and Y2 receptors are known to be expressed in the epithelium of human colon mucosa, only Y2 is expressed in the intrinsic primary afferent neurons of the mucosa that control motility, mucus secretion, and local blood flow [40,41]. Since we have identified luminal PYY (3–36) secretion and DPP4 activity in human colon mucosa, we can assume that the cleaved PYY (3–36) form is predominant to PYY (1–36) and that Y2 receptors in the lumen are preferentially activated, thus reducing gastric emptying.

In conclusion, our findings demonstrate that, under stimulation with meat peptone, PYY and GLP1 are secreted in the luminal compartment where they may have a direct impact on the lumen of human colon mucosa. GCGR expression in colon suggests that this receptor binds GLP1 (9–36) or glucagon to mediate the action of these hormones. Although PYY seems to be involved in the regulation of gastric emptying and GLP1 shows a diffuse expression of its classical receptor in many different cell types of the mucosa [42], further research is needed to shed light on the specific roles of these enterohormones in this organ. Fully understanding the role of enterohormones is crucial for obesity and diabetes, since they are among the main targets of the current therapies of these metabolic diseases.

## 4. Materials and Methods

### 4.1. Materials

Peptone from bovine meat, enzymatically digested (Cat. No: 70175, Sigma-Aldrich, Madrid, Spain), was used as a treatment and protease inhibitors were used in the media in the enterohormone secretion studies. The specific inhibitors and their working concentrations were: 10 µM amastatin (Enzo Life Sciences, Madrid, Spain), 100 KIU aprotinin (Sigma, Madrid, Spain), and 0.1% bovine serum albumin (BSA) fatty acid free. For cell culture experiments, DMEM (with 4.5 g L^−1^ glucose), L glutamine solution (200 mmol L^−1^ in 0.85% NaCl), penicillin-streptomycin mixture (10,000 U mL^−1^), and trypsin–EDTA solution (500 mg L^−1^ trypsin and 200 mg L^−1^ EDTA in Hank’s Balanced Salt Solution) were purchased from Lonza Verviers SPRL (Verviers, Belgium). Fetal bovine serum (FBS) was provided by Sigma–Aldrich Chemie (Steinheim, Germany). RPMI 1640 medium and HEPES buffer solution (1 mol L^−1^) were from GIBCO (New York, NY, USA). Phorbol-12-myristate-13-acetate (PMA) was provided by Quimigrancel (Barcelona, Spain). Different primary and secondary antibodies, listed in Appendix A, were used for Western Blot and immunofluorescence.

### 4.2. Obtaining Intestinal Mucosa Samples

Samples were obtained from the human ascending and descending colon and processed as previously described [43], using the healthy margin of the mucosae near the fragment that is excised from colon cancer patients who require colectomy. These procedures were conducted at the Hospital Universitari Joan XXIII in Tarragona, Spain. The subjects were aged between 52 and 78. The exclusion criteria were: alcohol intake above 30 g/day; body mass index above 40 kg/m^2^; use of drugs unrelated to metabolic syndrome treatment; the presence of intestinal malabsorptive or inflammatory bowel diseases; the presence of acute or chronic inflammatory or infectious disease; and the presence of neoplastic disease that was advanced or required pharmacological treatment. Appendix A shows the characteristics of the subjects who took part in the study. The samples were obtained during surgical treatment. Written consent has been obtained from each patient or subject after full explanation of the purpose and nature of all procedures used.

### 4.3. Enterohormone Secretion Study in Explants

Human colon mucosa was cut into sections using a biopsy punch 6 mm of diameter. After being washed for 15 min in KRB-D-Mannitol buffer saturated with 95% oxygen and 5% CO_2_, tissue segments were placed in pre-warmed (37 °C) KRB-d-Glucose buffer 0.1% DMSO with protease inhibitors and either meat peptone or vehicle. The samples were treated for 30 min in a humidified incubator at 37 °C, 95% O_2_, and 5% CO_2_. After 30 min of treatment, the whole volume was frozen and stored at −80 °C for enterohormone quantification.

### 4.4. Enterohormone Secretion Study in an Ussing Chamber

Human colon mucosa was cut into 1 cm × 1 cm sections. These sections were placed in an Ussing chamber with a 6 mm aperture (Dipl.-Ing. Muβler Scientific Instruments, Aachen, Germany). The mucosal preparations were stabilized for 15 min in a bath of 1.5 mL KRB-D-Mannitol in the apical chamber and 1.5 mL KRB-D-Glucose in the basolateral chamber. Next, baths in both chambers were replaced with KRB-D-Glucose, with protease inhibitors containing either meat peptone or vehicle. Bathing solutions were oxygenated and circulated in water-jacketed reservoirs maintained at 37 °C. After 30 min of treatment, the entire volumes of the apical and basolateral sides were frozen at −80°C for subsequent analysis of the enterohormones. Transepithelial electrical resistance (TEER) (ohm × cm^2^) was measured through short-circuit current using Ag-AgCl electrodes.

### 4.5. Enterohormone Quantification

The enterohormones in the intestinal media collected during the enterohormone secretion studies were assayed using commercial ELISA kits in accordance with the manufacturer’s instructions: Human PYY (3–36) was measured with fluorescence immunoassay (FEK-059-02, Phoenix Pharmaceuticals, Burlingame, CA, USA). Human total GLP-1 was measured using an ELISA kit (Cat No.: EZGLPT1-36k, Millipore, Burlington, MA, USA).

### 4.6. DPP4 Activity Measurements

DPP4 activity from human colon mucosa lysates was measured with a fluorimetric assay, using H-Gly-Pro-AMC HBr (Bachem, Bubendorf, Germany) as substrate. DPP-4 activity was normalized to total tissue weight.

### 4.7. Cell Culture

Caco-2 (HTB-37) and THP-1 (TIB-202) cells were obtained from the ATCC (American Tissue Culture Collection) and used between passages 19–23 and 2–24, respectively. Raji-B cells were kindly provided by Dr Juan Otero from the Immunology department of the Hospital Clínic (Barcelona, Spain). For details on cell culture, see Appendix A.

### 4.8. Immunofluorescence Staining

Immunofluorescence staining was performed in human mucosa samples and fixed human colonic cell culture models grown on porous membranes: CaCO_2_ cells monoculture, CaCO_2_ + THP1 macrophages co-culture and CaCO_2_ + THP1 + Raji B lymphocytes tri-culture. Samples were fixed in 4% paraformaldehyde in 0.1 m phosphate buffered saline (PBS) with a pH of 7.4 for 24 h and stored in 70% ethanol. After gradient dehydration, samples were embedded transversally to the cutting direction in paraffin blocks in order to ensure good visualization of the mucosa. Embedded tissues were cut into sections 5 μm thick that were fixed to polarized glass slides. Samples were rehydrated and stained, following the previously used method [44], with a few modifications. Antigens were retrieved from rehydrated samples with a basic antigen retrieval solution (10 mm Tris base, 1 mm EDTA solution, pH 9) at 90 °C for 20. The blocking solution used was 0.1% bovine serum albumin (BSA) in PBS for 20 min at RT. The anti-GLPR, anti-GCGR, and anti-actin primary antibodies and the appropriate fluorochrome-conjugated secondary antibodies were diluted in BSA 0.01% PBS and are specified in Appendix A. The double labelling was performed as follows: GLP1R and actin.

### 4.9. Statistical Analyses

The results are presented as mean ± SEM. Data were analyzed using XLSTAT 2020.1 (Addinsoft, Barcelona, Spain) statistical software. Statistical differences were assessed by subjecting the means of each treatment and the control to Student’s *t*-tests or one-way ANOVA tests when appropriate. Significance was accepted over 5%.

## Figures and Tables

**Figure 1 ijms-23-03523-f001:**
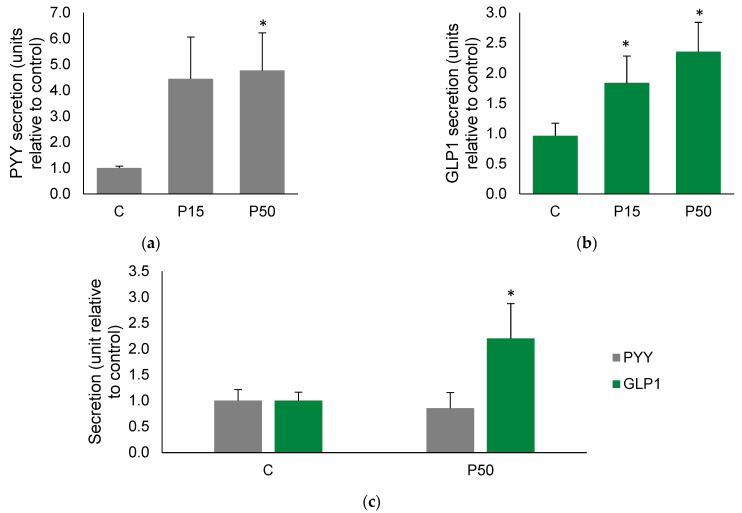
Enterohormone secretions from ex vivo intestinal segments of human colon. In grey, PYY (**a**) and, in green, GLP1 (**b**) secretion from free explants in untreated Control (C) after treatment with 15 and 50 mg/mL meat peptone (P15 and P50). (**c**) PYY (in grey) and GLP1 (in green) secretion in Ussing chambers in response to P50. Data are expressed as mean fold increase over basal secretion ± SEM in *n*= 9–13 experiments; * indicates significant differences (*p* < 0.05) between control and treated conditions.

**Figure 2 ijms-23-03523-f002:**
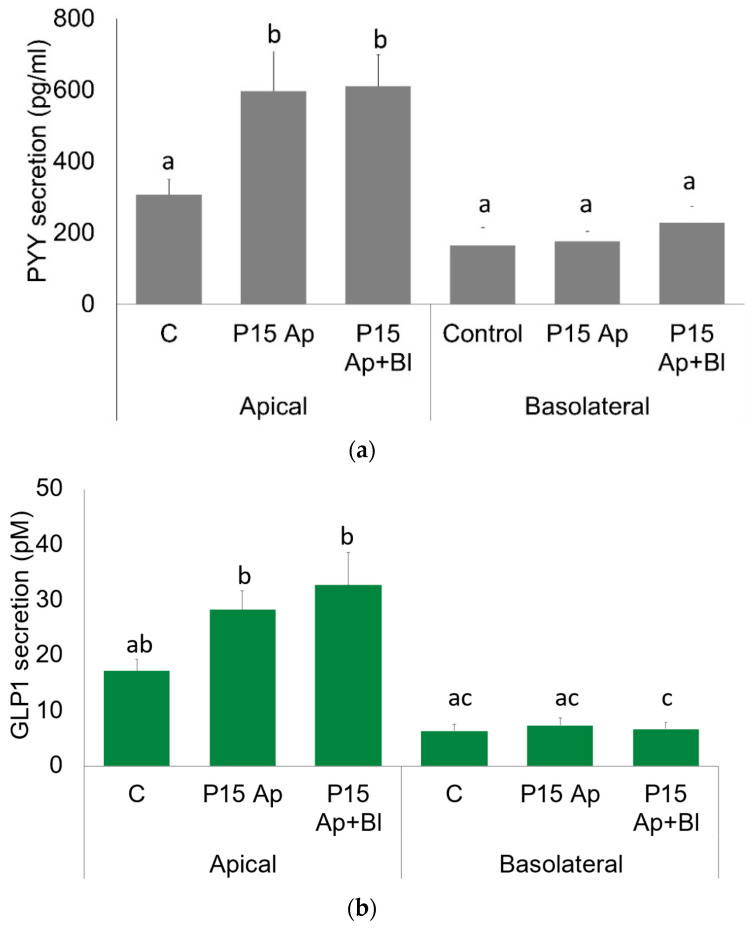
Apical and basolateral secretion of PYY and GLP1 in human colon. (**a**) In grey, PYY secretion from human colon in the apical and basolateral compartments of Ussing chambers in untreated control (C) and after apical (P15 Ap) or apical + basolateral (P15 Ap + Bl) treatment with 15 mg/mL meat peptone. (**b**) In green, GLP1 secretion from human colon in apical and basolateral compartments of Ussing chambers in untreated control (C) and after apical (P15 Ap) or apical + basolateral (P15 Ap + Bl) treatment with 15 mg/mL meat peptone. Data are expressed as mean ± SEM hormone secretion in *n*= 9–13 experiments; significantly different groups by a one-way ANOVA with a Tukey post-hoc test are indicated with a, b, and c.

**Figure 3 ijms-23-03523-f003:**
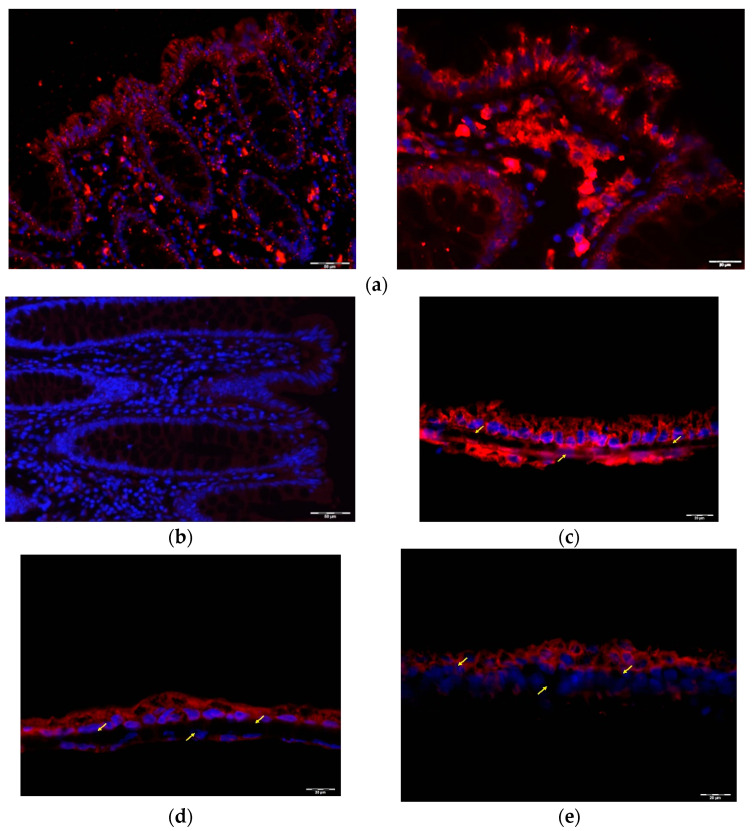
Immunofluorescence images of the human colon mucosa and colonic cell culture stained with GLP1R antibody. (**a**) GLP1R (red) positive cells in the colon epithelium (dotted line) and in the colon lamina propria (circle)**,** GLP1R negative cells in the crypts of Lieberkühn (arrow). Representative of *n*= 4 preparations. (**b**) Negative control of human colon mucosa, stained only with secondary antibody. (**c**) GLP1R (red) positive CaCO_2_ polarized colonocytes. (**d**) GLP1R (red) positive CaCO_2_ + THP1 co-culture. (**e**) GLP1R (red) positive CaCO_2_ + THP1 + Raji B tri-culture. Representative of *n*= 4 preparations. Membranes used as support for cell growth are marked with yellow arrows (3C, 3D, and 3E). Nuclei were stained with DAPI (blue).

**Figure 4 ijms-23-03523-f004:**
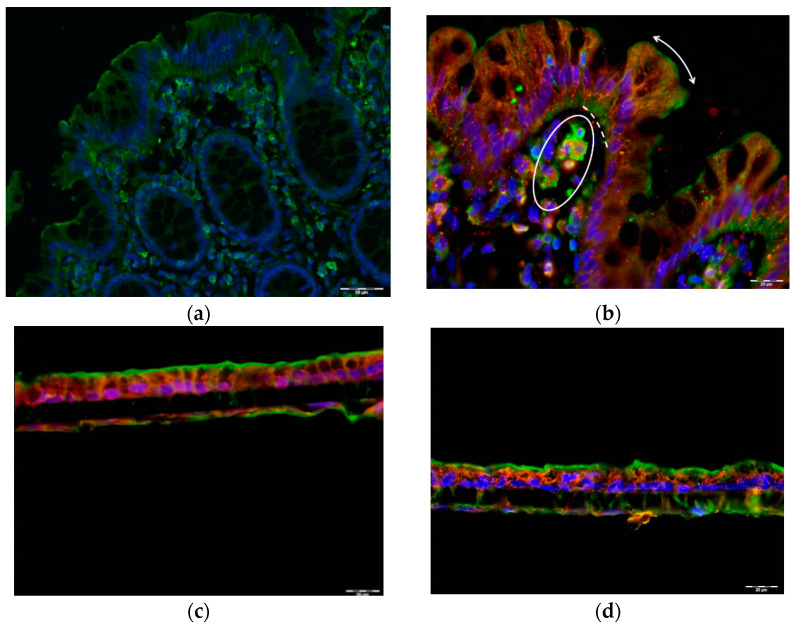
GLP1R localization in human colon mucosa and in in vitro coculture CaCO_2_ cell models. (**a**) Actin (green) staining of human colon mucosa. Apical actin filaments in the brush border are indicated with a two-sided arrow. (**b**) Actin (green) and GLP1R (red) co-staining of the epithelium (dotted line) and lamina propria (circle) in human colon mucosa. Apical actin filaments in the brush border are indicated with a two-sided arrow. (**c**) Actin (green) and GLP1R (red) co-staining of CaCO_2_ + THP1 co-culture. (**d**) Actin (green) and GLP1R (red) co-staining of CaCO_2_ + THP1 + Raji B tri-culture. Representative of *n*= 4 preparations. Nuclei were stained with DAPI (blue).

**Figure 5 ijms-23-03523-f005:**
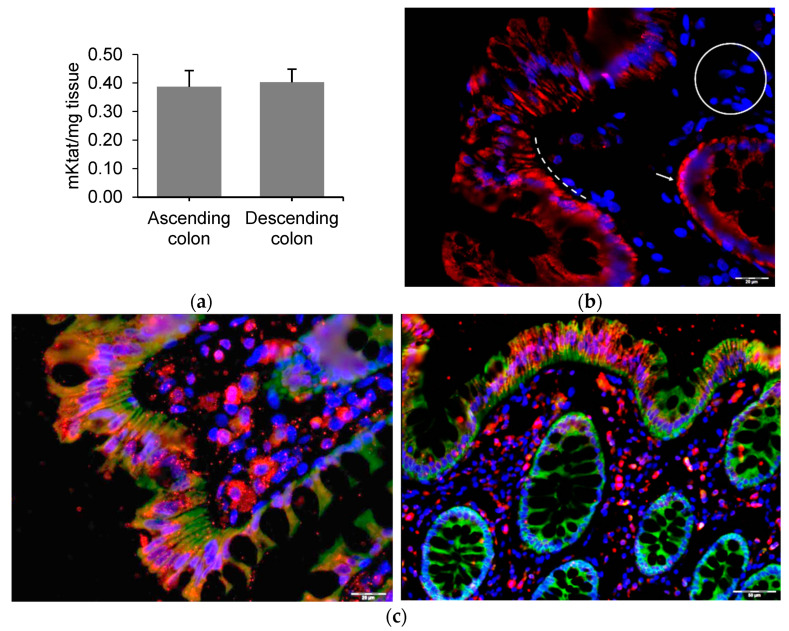
DPP-IV activity and presence of glucagon receptor in human colon mucosa. (**a**) DPP-IV activity in lysates from human ascending and descending colon mucosa. Data are expressed as mean ± SEM of *n*= 4 preparations. (**b**) Glucagon receptor (GCGR) positive (red) cells in human colon epithelium (dotted line) and crypts of Lieberkühn (arrow) and negative cells in the lamina propria (circle); representative of *n*= 4 preparations. (**c**) GCGR (green) and GLP1R (red) co-staining of human colon mucosa. Representative of *n*= 4 preparations from independent models. Nuclei were stained with DAPI (blue).

## Data Availability

Not applicable.

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
