# Peer review of "GLP1 Exerts Paracrine Activity in the Intestinal Lumen of Human Colon"

_ijms, 2022, doi:10.3390/ijms23073523_

Round 1
Reviewer 1 Report
In the current manuscript “GLP1 exerts paracrine activity in the intestinal lumen of human colon” by Grau-bove et al. using human colon ex-vivo culture and immunofluorescence staining showed that the PYY and GLP1 stimulated by meat peptone. Showed the expression of receptors GLP1R and GCCR.
Comments
- In figure 3a, GLP1R staining along with colonocytes or goblet cell specific markers could be stained to confirm the authors claim that it expressed in colonocytes only. Immunofluorescence staining in presence and absence of meat peptone should be performed for GLP1R.
- In figure 4, although authors performed and validated GLP1R expression in by using actin but colonocyte specific markers also used. In figure 4d and e, GLP1R expression appears to be basolateral.
- In figure 4c, as author claims no mucin expression by PAS staining and observed only in goblet cells. To observe mucin expression the tissue sections should not be washed. Therefore it is obvious not to see mucin expression. Colonocyte specific marker staining should be performed.
- In presence and absence of meat peptone the receptor (GLP1R and glucagon receptor) staining and expression analysis by RT-PCR from lamina propria and epithelial cells from tissue biopsies should be performed.
- Scale bars are missing on the microscope images.
- Indicate how the normalization for the biopsy tissue size differences were not the factor of differences in secretion of enterohormones in ELISA.
- 50 mg/ml meat peptone treatment does not affect the overall viability or integrity of the biopsy tissue.
Author Response
Dear reviewer,
We are very thankful for your comments on our manuscript. We believe that following your suggestions we have increased the quality of it. We have made modifications according to your comments (as explained below, point by point) and to the other reviewer’s suggestions, as well as some English changes.
- In figure 3a, GLP1R staining along with colonocytes or goblet cell specific markers could be stained to confirm the authors claim that it expressed in colonocytes only. Immunofluorescence staining in presence and absence of meat peptone should be performed for GLP1R.
We agree with you that it would be interesting to decipher which specific cell types in the human colon epithelium are positive for GLP1R. However, the aim of our study was to investigate if GLP1R is expressed in the colon epithelium to argue that GLP1 apical secretion in the intestine can be detected.
Since we have not used specific markers for the different cell types, we understand that we cannot state that the signal we observe is in these specific cell types. Nevertheless, the results in Caco2 cells and the wide positivity in the colonic epithelial cells strongly suggest a positivity in the colonocytes. For this reason, we said in lines 132 and 281 that the GLP1R positivity in the colon epithelium only suggests a GLP1R positivity in colonocytes, instead of stating it. Nevertheless, in some other parts of the manuscript we mentioned GLP1R-positive colonocytes, so we have revised it to replace these affirmations for GLP1R-positive colonic epithelial cells (lines 157, 162, 176 and 202).
We would also be very interested in studying the behaviour of GLP1R and also GCGR in the presence of peptone since we believe that future research has to be focused on understanding the function of these receptors expressed in the colon epithelium. Unfortunately, it is impossible for us to obtain and treat new tissue and perform new immunostainings within the given 10 days to respond to the reviews.
- In figure 4, although authors performed and validated GLP1R expression in by using actin but colonocyte specific markers also used. In figure 4d and e, GLP1R expression appears to be basolateral.
We do not fully comprehend the first sentence of this point, we guessed that you meant that cell-specific markers should have been used in figure 4, as you suggested in the previous point. As we responded to point 1, although we agree that it would be interesting to address which specific cell types express the GLP1R, we have already fulfilled our aim of finding the GLP1R signal in the epithelial cells of the human colon. In old Figure 4d and 4e (now Figure 4c and 4d respectively), we observe that GLP1R is expressed in the inside of the epithelial cells, and not in the apical or basolateral membranes. We have clarified this finding in the text (line 175).
- In figure 4c, as author claims no mucin expression by PAS staining and observed only in goblet cells. To observe mucin expression the tissue sections should not be washed. Therefore it is obvious not to see mucin expression. Colonocyte specific marker staining should be performed.
We agree with you that mucins are not expected in the surface of the colon having washed the tissue. Therefore, we have removed this figure from the manuscript, but we have decided to keep it in the supplementary material file (sup. fig. 3), because it is still of interest to identify the goblet cells in the crypts from which the mucins were not washed off. Regarding colonocyte identification, as we have previously said, we have reviewed the text to say that the findings suggest a GLP1R positivity in these cells.
- In presence and absence of meat peptone the receptor (GLP1R and glucagon receptor) staining and expression analysis by RT-PCR from lamina propria and epithelial cells from tissue biopsies should be performed.
As answered in point 1, we believe that future research should address the expression of GLP1R and GCGR in stimulated conditions but, regrettably, we do not hold the possibility to perform this study within the current review time.
- Scale bars are missing on the microscope images.
Thank you for pointing this error out, we have added the missing scale bars in figures 3c-e and 4a.
- Indicate how the normalization for the biopsy tissue size differences were not the factor of differences in secretion of enterohormones in ELISA.
As indicated in the method section, all tissue biopsies used for enterohormone secretion studies were of the same dimensions, as they were cut out from the fresh colon samples using a 6 mm diameter round punch. Therefore, normalization by tissue weight was not necessary in this study as it was for DPP4 activity analysis, for example, in which portions of tissue were obtained from breaking frozen colon mucosal samples.
- 50 mg/ml meat peptone treatment does not affect the overall viability or integrity of the biopsy tissue.
We agree with you in that 50 mg/ml meat peptone does not affect viability. However, as we saw that it reduced TEER (transepithelial electric resistance) of the tissue, we selected a lower dose (15 mg/ml) for subsequent analysis. We have revised the text to clarify that 50 mg/ml meat peptone does not damage the tissue (line 84).
Reviewer 2 Report
The study tries to investigate the potential function though the excretion of GLP 1 receptor by GLP1 secreted in human colon. The conclusions drawn from the results are appropriate. However, statistical tests and significance should be included in the manuscript and the data from the immunofluorescence experiments should be extended. Below are comments to enhance the manuscript:
1) The right statistical test should be used. In the method section it is noted that an unpaired t-test was used for testing statistical differences. However, the data in figure 1 and figure 2 requires ONEWAY ANOVA or TWOWAY ANOVA testing, respectively to conclude on significant differences between groups.
2) In Supplementary Figure 2, although the variation of GLP1R expression seem to have high variation on western blot, quantified results is not high variation. how many samples were included in quantified data?
3) Authors showed that GLP1R was expressed using immunofluorescence in human colon and colonic cell in figure 3. However, it does not show the part about whether the location is correct using co-staining as like figure 4 and 5. Author should make sure to identify location for example, immunofluorescence use anti-lgr5, which is positive marker of crypt base columnar cell (Cell. 2013 Jul 18;154(2):274-84.).
4) In Supplementary table 1, authors used anti-actin (goat). But anti-goat 2nd antibody was not there in table.
Round 2
Reviewer 1 Report
Authors addressed the raised concerns.